# Method for Dynamic Prediction of Oxygen Demand in Steelmaking Process Based on BOF Technology

**Kaitian Zhang [1], Zhong Zheng [1,*], Liu Zhang [1], Yu Liu [1] and Sujun Chen [1,2]**

[1] College of Materials Science and Engineering, Chongqing University, Chongqing 400045, China
[2] Shougang Jingtang United Iron and Steel Co., Ltd., Tangshan 063299, China
* Correspondence: zhengzh@cqu.edu.cn

**Abstract:** Oxygen is an important energy medium in the steelmaking process. The accurate dynamic prediction of oxygen demand is needed to guarantee molten steel quality, improve the production rhythm, and promote the collaborative optimization of production and energy. In this work, a analysis of the mechanism and of industrial big data was undertaken, and we found that the characteristic factors of Basic Oxygen Furnace (BOF) oxygen consumption were different in different modes, such as duplex dephosphorization, duplex decarbonization, and the traditional mode. Based on this, a dynamic-prediction modeling method for BOF oxygen demand considering mode classification is proposed. According to the characteristics of BOF production organization, a control module based on dynamic adaptions of the production plan was researched to realize the recalculation of the model predictions. A simulation test on industrial data revealed that the average relative error of the model in each BOF mode was less than 5% and the mean absolute error was about 450 m$^3$. Moreover, an accurate 30-minute-in-advance prediction of dynamic oxygen demand was realized. This paper provides the method support and basis for the long-term demand planning of the static balance and the short-term real-time scheduling of the dynamic balance of oxygen.

**Keywords:** steelmaking; basic oxygen furnace mode; oxygen demand; dynamic prediction; big data

## 1. Introduction

Steel is the world's most important engineering and construction material [1]. Low-carbon and intelligent are the trends of the steel industry [2]. Under the strategy of Industry 4.0, Europe has begun to develop hydrogen metallurgy, direct-reduction ironmaking, electric arc furnace, and digitization to provide technical support for the development of the steel industry [3]. However, traditional Basic Oxygen Furnaces (BOFs) were still responsible for approximately 70% of steel production worldwide in 2021 [4].

During the BOF process, scrap and hot metal are charged into a BOF, and an oxygen jet is injected at supersonic speed from the top through the lance onto the surface of the metal bath. As carbon, silicon, manganese, phosphorus, sulfur, and other elements are oxidized and removed, the hot metal is heated and smelted into molten steel. With the improvement in the requirements for the composition and temperature at the end of BOFs, the functions of traditional BOFs, such as "three desorption" and heating, have been gradually decomposed into dephosphorization–decarbonization duplex technology [5,6]. This technology not only reduces the cost but also brings challenges to the energy guarantee and production organization of the BOF steelmaking process [7,8].

Oxygen is one of the most important energy sources in the BOF steelmaking process and plays an important role in the production rhythm and component temperature control [9,10]. Due to the function of different BOF modes, they have different oxygen demands. At the same time, intermittent production and multi-ladle superposition organization lead to a large fluctuation in oxygen demand, which contradicts the continuous and stable production of oxygen generators, leading to a mismatch in oxygen supply and

demand [11,12]. When the supply is less than the demand, oxygen cannot meet the requirements for normal BOF blowing, affecting the production rhythm. On the other hand, when the supply exceeds the demand, to ensure the safety pressure of the oxygen pipe, excess oxygen has to be dispersed, resulting in economic losses. Therefore, it is very important to accurately predict oxygen demand in the BOF steelmaking process [13–16].

At present, the oxygen prediction models for BOFs mainly include static prediction and dynamic prediction. Dynamic prediction mainly depends on auxiliary monitoring technology and gas analysis technology to revise a BOF's terminal point in real time [17–19]. However, most of the dynamic prediction models aim at a BOF's endpoint to optimize the process operation, which results in oxygen demand prediction only serving as the optimization means of composition and temperature rather than providing the scheduling of the oxygen pipe network balance. Research on oxygen balance, however, is still focused on the energy side, which is mainly static balance. For example, Ruuska et al. [20] built a multivariate coupled model based on mass balance to predict the distribution of chemical elements in BOFs. Daniela et al. [21] established a model based on thermodynamics and kinetics to simulate liquid steel composition, oxygen consumption, and other factors during the BOF process. However, the thermodynamic and kinetic conditions of the BOF process are complicated, leading to difficulties in the theoretical calculation of reactions such as element oxidation, auxiliary material melting, and furnace lining erosion. With its fast response speed and high prediction accuracy, the data-driven intelligent model has gradually become a new method to improve the prediction accuracy and efficiency of BOF modeling. For example, Liu et al. [22] proposed an effective mechanical-data fusion modeling method for an energy (physical heat, reaction heat, and consumption heat of molten steel)-informed restricted Boltzmann machine to accurately track the BOF process and dynamically optimize operation control. Peng et al. [23] optimized the structural parameters of a Support Vector Machine (SVM) with an optimization algorithm and established a prediction model of oxygen consumption in BOFs. Jiang et al. [24] used the mixed model of multiple linear regression (MLR) and Gaussian process regression (GPR) to predict the oxygen consumption of BOFs. Nevertheless, the above models all improved the model by optimizing the algorithm instead of employing theoretical interpretation, resulting in insufficient applicability. And the application effect of the same algorithm in different BOF modes was quite different.

Moreover, oxygen demand in the BOF steelmaking process fluctuates greatly due to the BOF modes and complex organization. The adjustment based on the static balance of the oxygen network has a time delay, leading to imbalances in oxygen supply and demand and affecting the production stability. Optimal oxygen scheduling based on the collaboration between the production system and the energy system has become a new study direction. In this field, Zhang et al. [25], Kong et al. [26], and Xu et al. [27] considered the oxygen system characteristics of production, storage, buffering, and consumption and established an optimal oxygen scheduling model based on the operational constraints and scheduling rules of steelmaking. They made some useful attempts to design production-energy multi-system collaborative modeling methods. However, these oxygen models did not accurately predict the oxygen demand in each period of the BOF process in combination with the production plan and the BOF mode. As a result, the models could only predict statically rather than dynamically under the real industrial condition where the BOF production plan changed frequently. And it was difficult to provide effective support for the strategy to ensure the optimal scheduling of the oxygen pipe network to establish a dynamic balance.

Consequently, this work combined the BOF steelmaking mechanism and industrial big data to analyze the characteristics of oxygen consumption under different BOF modes. The oxygen demand prediction models were established by a neural network algorithm for duplex dephosphorization, duplex decarbonization, and traditional mode. On this basis, considering the production organization and the dynamic scheduling, a dynamic optimization module was developed using the data dynamic interaction technology to realize the dynamic prediction and correction of the real-time oxygen demand, which would

support the stable BOF steelmaking rhythm and the oxygen optimization scheduling to establish a pipe network dynamic balance.

## 2. Materials and Methods

The subject of this work was to establish a prediction model of oxygen demand of converter based on the characteristics of duplex dephosphorization, duplex decarburization, and traditional modes, to provide decision-making support for oxygen scheduling. The practical section was divided into three parts. The first part was the theoretical analysis which described the BOF oxygen demand characteristics combined with the mechanism and historical data analysis. The second part described the process of establishing the prediction model and its operating logic. The last part described the industrial data preprocessing methods. All the data were from a Chinese steelmaking plant. About 50,000 heats were collected.

### 2.1. Analysis of BOF Oxygen Demand Characteristics Combined with the Mechanism and Data

2.1.1. Analysis of Oxygen Consumption Mechanism in BOF

In the process of BOF steelmaking, the oxygen was mainly from oxygen lance ($m_{O_2, \text{blow}}$) and carried by auxiliary materials ($\sum\limits_{\text{ingredients}} m_{[O], j}$) such as iron ore, lime, dolomite, etc. The consumption of oxygen was mainly divided into three categories: $\sum\limits_{\text{oxidation}} m_{[O], i}$, the oxidization of elements in hot metal such as [Fe], [C], [Si], [Mn], [P]; $m_{[O], \text{brasque}}$, the oxidization of carbon-containing materials in furnace lining; and $m_{O_2, \text{dissipation}}$, the excess oxygen taken away by furnace gas.

The auxiliary materials containing oxygen mainly included lime, iron ore, dolomite. According to their main components, they could be simplified as (CaO), ($FeO \cdot Fe_2O_3$), ($CaCO_3 \cdot MgCO_3$). Then,

$$\sum_{\text{ingredients}} m_{[O], j} = M_O \cdot \frac{m_{\text{lime}}}{M_{CaO}} + M_O \cdot 4 \frac{m_{\text{ironstone}}}{M_{FeO \cdot Fe_2O_3}} + M_O \cdot 6 \frac{m_{\text{dolomite}}}{M_{CaCO_3 \cdot MgCO_3}} \tag{1}$$

where $m_i$ is the amount of each auxiliary material; $M_i$ is the molar mass of each substance.

In the hot metal, the oxidation of [Fe] produces (FeO) and ($Fe_2O_3$). The oxidation of [C] produces {CO} and {$CO_2$}. The oxidation of [Si] produces ($SiO_2$). The oxidation of [Mn] produces (MnO). The oxidation of [P] produces ($P_2O_5$). As a consequence, $\sum\limits_{\text{oxidation}} m_{[O], i}$ could be simplified by Equation (2).

$$\sum_{\text{oxidation}} m_{[O], i} = M_O \cdot \frac{\Delta m_{[Fe, FeO]}}{M_{Fe}} + M_O \cdot 1.5 \frac{\Delta m_{[Fe, Fe_2O_3]}}{M_{Fe}} + M_O \cdot \frac{\Delta m_{[C, CO]}}{M_C} + M_O \cdot 2 \frac{\Delta m_{[C, CO_2]}}{M_C} +$$
$$M_O \cdot 2 \frac{\Delta m_{[Si]}}{M_{Si}} + M_O \cdot \frac{\Delta m_{[Mn]}}{M_{Mn}} + M_O \cdot 2.5 \frac{\Delta m_{[P]}}{M_P} \tag{2}$$

The oxygen consumed by the carbon in the BOF lining material could be simplified as:

$$m_{[O], \text{brasque}} = M_O \cdot \frac{m_{[C, CO]}}{M_C} + M_O \cdot 2 \frac{m_{[C, CO_2]}}{M_C} \tag{3}$$

The oxygen taken away by furnace gas needed to be detected by the flue gas of the BOF.

According to the above analysis, oxygen consumption in the BOF steelmaking process could be simplified as Equation (4). In this equation, $k_i$ is the simplified expression of a series of coefficients of the complex relationship between each element of hot metal and oxygen consumption; $k_j$ is the simplified coefficients of the complex relationship between auxiliary material, such as lime, iron ore, dolomite, and oxygen input; and $K$ is the empirical coefficient of furnace lining oxidization and excess oxygen dissipation. Due to the complicated temperature and equipment conditions, it was difficult and costly to accurately

measure the above coefficients in different production environments. For example, to calculate the oxidation coefficient of [C] (kc), the furnace gas detection equipment was needed to analyze the ratio of decarbonized products CO and $CO_2$, which had a high cost and limited accuracy. It was also necessary to compare the furnace lining erosion before and after production, which made it difficult to establish the fast production rhythm. However, for most variables, such as the amount, the content, or the temperature of hot metal or molten steel, the BOF steelmaking process was equipped with relatively mature detection methods, which could accurately obtain the real industrial production data. Therefore, among the influencing factors of the BOF oxygen consumption listed in Equation (4), the mature acquisition conditions in industrial sites such as $m_{iron}$, $[i\%]_{iron}$, $m_{steel}$, $[i\%]_{steel}$, and $T$ were preliminarily selected as the oxygen demand prediction modeling factors.

$$m_{O_2,\,blow} = \sum_{\substack{oxidation}}^{i=Fe,C,Si,\cdots} k_i \cdot \left( m_{hot\,metal} \cdot [i\%]_{hot\,metal} + m_{scrap} \cdot [i\%]_{scrap} - m_{steel} \cdot [i\%]_{steel} \right)/100 - \sum_{\substack{ingredients}}^{j=CaO,FeO,\cdots} k_j \cdot m_{(j)} + K \quad (4)$$

### 2.1.2. Analysis of BOF Mode Based on Industrial Big Data

The statistics of oxygen consumption rates of more than 25,000 heats in a Chinese steelmaking plant is shown in Figure 1. According to the function, the BOF process could be divided into three modes: duplex dephosphorization (I), duplex decarbonization (II) and traditional mode (III). Mode I was mainly used to complete the de-[P] task and transfer hot metal into semi-steel. Mode II was to transfer the semi-steel from Mode I into qualified molten steel, which mainly completed the de-[C], de-[S], and temperature-rising tasks. Mode III directly transferred the hot metal into molten steel in a BOF to complete the de-[P], de-[C], de-[S], and temperature-rising tasks at the same time. Figure 1 shows that the oxygen consumption was concentrated around 4000 m³, 11,000 m³, and 14,000 m³, which corresponded to BOF Mode I, II, and III. From the perspective of hot metal input conditions, as shown in Figure 2a, both de-[C] and tonnage were positively correlated with oxygen consumption, and this relationship was consistent in the three BOF modes. However, the influences of temperature, as shown in Figure 2b, differed in the three BOF modes. For Mode I duplex de-[P], the BOF input was hot metal with large temperature fluctuations while the output was semi-steel with a stable temperature. A higher temperature in BOF was not conducive to the thermodynamic conditions for de-[P], so its oxygen consumption was lower than that at a lower temperature of hot metal, showing that temperature and oxygen consumption were negatively correlated. For Mode II duplex de-[C], the BOF input was semi-steel and the output was molten steel with different end-point temperatures. Since de-[C] is an exothermic reaction, the higher terminal temperature is, the more thoroughly the reaction proceeds. Therefore, the temperature was positively correlated with oxygen consumption. For Mode III, the thermodynamic and kinetic conditions were more complex, so there was no obvious rule of temperature and oxygen consumption.

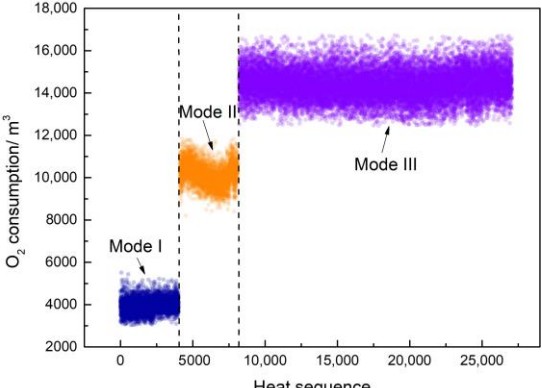

**Figure 1.** Statistics of oxygen consumption in BOF for different modes.

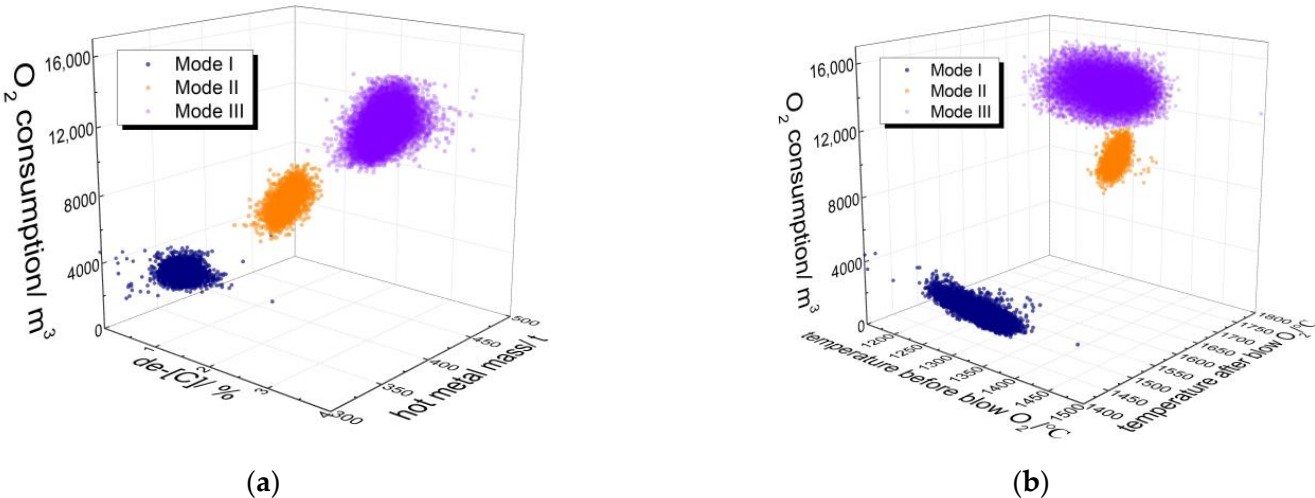

(**a**)                                                                      (**b**)

**Figure 2.** Statistics of oxygen consumption in BOF: (**a**) hot metal conditions; (**b**) temperature conditions.

The relationship between the main oxidation in hot metal and oxygen consumption is shown in Figure 3. Under different BOF modes, there were different influence rules between each element and oxygen consumption. For [Si], it was usually oxidized rapidly in the early stage of BOF steelmaking, so there was a significant positive correlation with oxygen consumption in both Mode I and Mode III. However, Mode II was the de-[C] stage of the duplex, and [Si] was mostly oxidized in the former de-[P] stage. Therefore, [Si] in Mode II had no obvious correlation with oxygen consumption. Similar to [Si], [Mn] was oxidized rapidly in the early stage. However, due to the increase in temperature and slag basicity at the later period, Mn reversion occured. The correlations between Mn and $O_2$ consumption under the three modes were not obvious. [P] was mainly oxidized by (FeO) and fixed by (CaO) as calcium phosphate into the slag, and this reaction was an exothermic reaction. In Mode I, the temperature was relatively low (about 1450 °C), which resulted in a slow rate of decarbonization. The [Fe] oxidation led to an increase in (FeO) content in the slag, which is conducive to dephosphorization. Therefore, Mode I had an obvious correlation with de-[P]. While the decarbonization reaction was intense in mode II and III, they had no obvious correlation. For [S], due to its low content, there were no obvious correlations between the three modes.

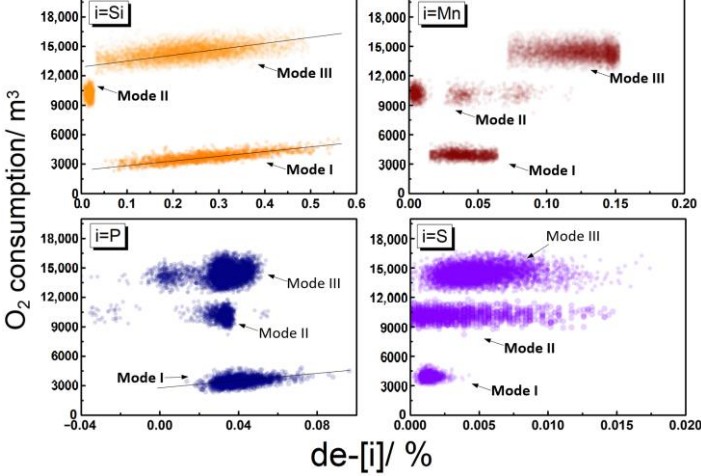

**Figure 3.** The relationship between the main oxidation in hot metal and the oxygen consumption.

In conclusion, different BOF modes had different effects on oxygen consumption due to different input materials, operations, functions, and targets. To achieve an accurate

prediction of oxygen consumption, it is necessary to construct different prediction models according to BOF mode, and to further refine corresponding characteristic factors and their influence rules on oxygen consumption to complete the modeling. For example, according to a correlation analysis of industrial data, the hot metal tonnage was a modeling factor for Mode I and II because it was an input to Mode I and II and had a clear correlation with oxygen consumption. For semi-steel weight, there was a correlation with oxygen consumption in Mode II, so this was a modeling factor in Mode II. Similarly, the corresponding characteristic factors of each mode can be obtained, as shown in Table 1. In addition, in the real industrial BOF steelmaking process, there could be one or more BOFs working at the same time based on the production plan, which led to dramatic fluctuations in oxygen flow demand. Therefore, in addition to the characteristic factors of the furnace oxygen consumption, the time characteristics should also be included to realize the real-time oxygen consumption flow prediction through the superposition of each furnace oxygen consumption.

**Table 1.** Characteristic factors of oxygen consumption in BOF.

| Characteristic Factors | Mode I | Mode II | Mode III | Unit |
|---|---|---|---|---|
| Hot metal tonnage | √ | | √ | t |
| Semi-steel tonnage | | √ | | t |
| Hot metal temperature | √ | | | °C |
| Semi-steel temperature | | √ | | |
| De-[C] | √ | √ | √ | % |
| De-[Si] | √ | | √ | % |
| De-[P] | √ | | | % |
| Start oxygen blowing time | √ | √ | √ | hh:mm:ss |
| End oxygen blowing time | √ | √ | √ | hh:mm:ss |

*2.2. Prediction Modeling of Oxygen Consumption in BOF*

2.2.1. Model Theory

As shown in Figure 4, the consumption curve of oxygen flow for a single BOF consists of a rising stage, a stable stage, and a falling stage. However, in the real production process, there were still several BOFs working together with different modes. Therefore, this work intended to establish a model to predict the oxygen consumption flow of a single BOF and the key time nodes of oxygen blowing, and then calculate the oxygen consumption flow of the steelmaking process by accumulating the oxygen consumption flow of each BOF.

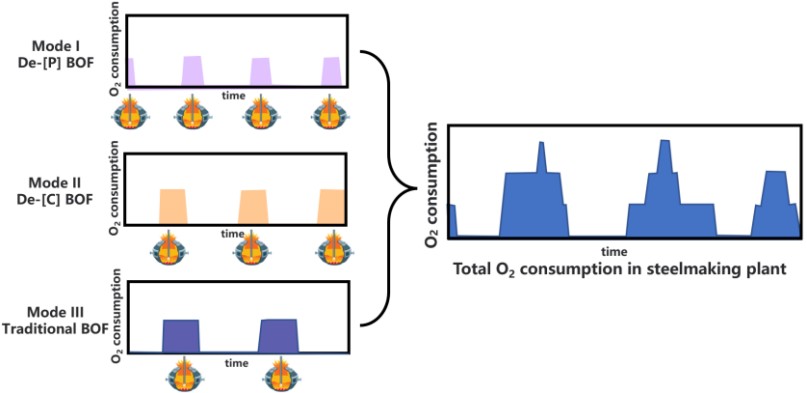

**Figure 4.** The relationship between the oxygen consumption of single BOF and that of steelmaking process.

For a single BOF $i$, the piecewise function of oxygen consumption flow $Q_{O_2,i}^t$ during the working process is shown in Equation (5). In the industrial BOF steelmaking process, the constant pressure operation was mostly adopted for oxygen lance, so it could be assumed

that the oxygen flow rate of each BOF in the stable blowing stage was constant, and the oxygen flow rate in the rising stage and the falling stage changed linearly. According to the industrial data, the average time in the rising stage ($t_1$) was 1.5 min, and that in the falling stage($t_2$) was 0.5 min. Similar to the calculation of trapezoid area, the $S_i^t$ could be calculated by oxygen consumption volume in BOF $V_{O_2,i}^t$ and oxygen blowing time $T_i$ according to Equation (6). The whole steelmaking process involving the production of multiple BOFs at the same time could be obtained by superposition of the oxygen flow rate of a single BOF according to the production plan. As shown in Equation (7), the oxygen flow demand $Q_{O_2,\text{total}}^t$ at the time of $t$ was the sum of the oxygen flow $Q_{O_2,i}^t$ consumed by all BOFs being produced at that time.

$$Q_{O_2,i}^t = \begin{cases} \frac{S_i^t}{t_1} \cdot (t - ost_i), & ost_i \leq t \leq ost_i + t_1 \\ S_i^t, & ost_i + t_1 \leq t \leq oet_i - t_2 \\ \frac{S_i^t}{t_2} \cdot (oet_i - t), & oet_i - t_2 \leq t \leq oet_i \\ 0, & \text{else} \end{cases} \tag{5}$$

where $ost_i$ and $oet_i$ are the start and end blowing time for $BOF_i$, respectively; $S_i^t$ is the oxygen flow rate at the stable stage, m$^3$/min; $t_1$, $t_2$ represent the average time in the rising stage and the falling stage respectively, min.

$$V_{O_2,i}^t = \frac{1}{2}(T_i^t + T_i^t - 1.5 - 0.5) \cdot S_i^t = (T_i^t - 1) \cdot S_i^t \tag{6}$$

$$Q_{O_2,\text{total}}^t = \sum_i Q_{O_2,i}^t \tag{7}$$

### 2.2.2. Model Algorithm Framework

In this work, a BP neural network model with double hidden layers was adopted, as shown in Figure 5. A BOF mode recognition algorithm was added before the main body of the model to determine the input variables according to the BOF mode. First, the BOF mode was determined according to the plan data; then, the corresponding characteristic variables were extracted according to the BOF mode as the input of the BP neural network model, and the model output the prediction of the oxygen consumption of the furnace; finally, the oxygen consumption of the furnace was converted into the oxygen flow rate according to Formulas (5)–(7). The input was the important process parameters that affected oxygen consumption, and the output was the $V_{O_2,i}^t$. Moreover, the $Q_{O_2,i}^t$ was obtained according to the oxygen-blowing duration. Finally, the multi-furnace oxygen blowing flow rate was superimposed according to the time needed to obtain the total oxygen consumption flow of the steelmaking process in real-time. After repeated optimization, the number of nodes in the hidden layer of the model was 20, the initial weight and threshold were randomly assigned between (0, 0.6) and (0~1), and the learning rate was 0.01.

The dynamic operation framework of the model included the parallel thread of prediction, clock, and training to improve the efficiency of the model. At the same time, a database was built to realize dynamic data interaction, as shown in Figure 6. The prediction thread was mainly used for loading the training model and outputting the results according to the trigger condition. The training thread was mainly responsible for completing the model training and storing it according to the trigger condition. The clock thread was the dynamic operation core of the model, which was used to store information such as the real clock, production schedule change mark, prediction and training trigger mark, etc. For example, when the clock ran to a certain time, the pre-made BOF production plan was changed due to the disturbance, the change flag of the production plan was triggered, and the model sample was updated. Finally, the model prediction thread updated the results to realize the dynamic prediction.

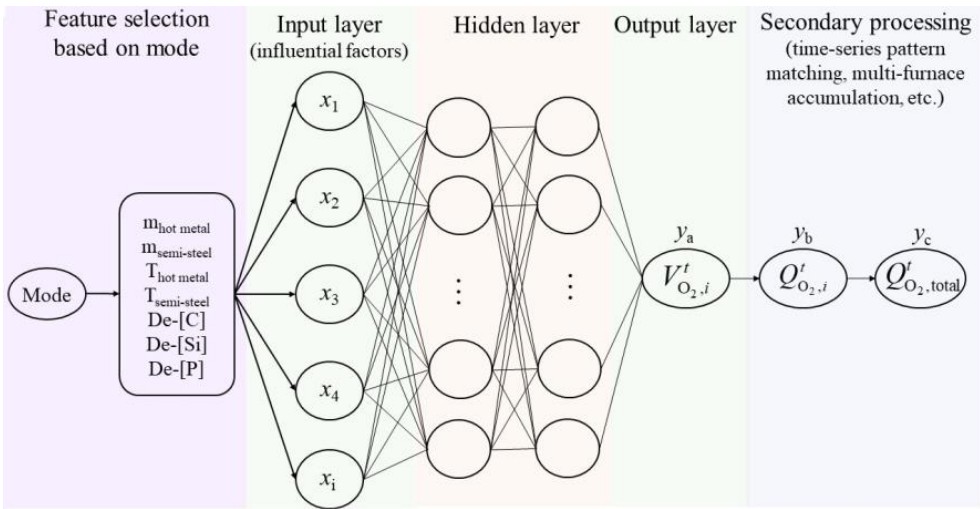

**Figure 5.** The framework of the prediction model for oxygen consumption of BOF.

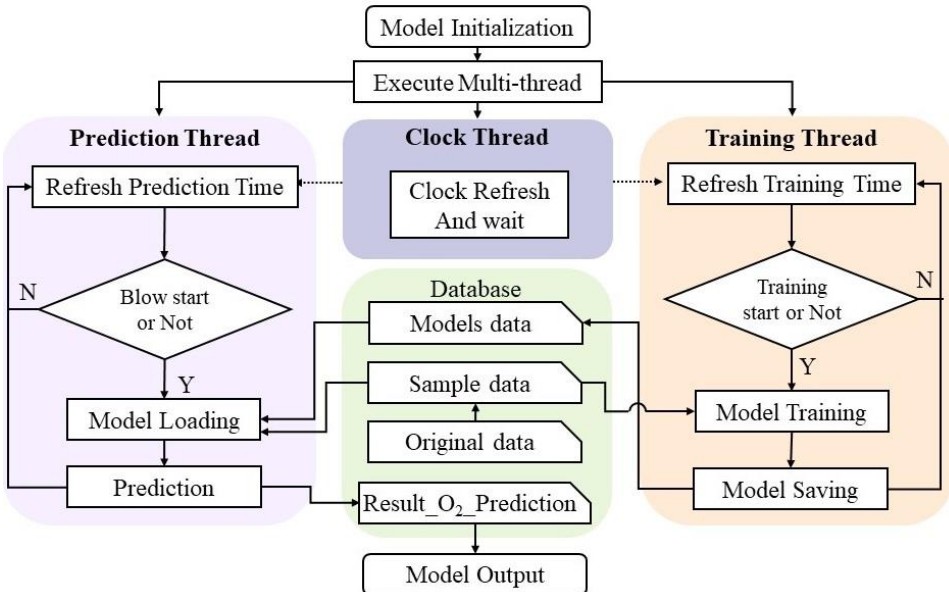

**Figure 6.** The dynamic operation framework of the model.

*2.3. The Establishment of Model Samples from Plan Data by Historical Big Data Preprocessing Method*

It can be seen from Table 1 that the BOF mode, information of hot metal/semi-steel (tonnage, component, temperature), information of the BOF terminal (component, temperature), and time information (start/end time of blowing) was required to obtain a complete sample of the prediction model. The above data were recorded in the performance data after production. However, the plan data before production generally included only station, BOF number, steel grade, start/end time, etc., which were quite different from the data items required by the model sample. Therefore, it is necessary to preprocess the production plan data to obtain reliable model learning samples.

2.3.1. Sample of Component and Temperature Characteristics

To calculate the de-[C], de-[Si], de-[P], and *T*, the real initial composition and temperature of hot metal/semi-steel and that at the BOF terminal must be obtained. Although there were no component and temperature data in the plan data, the hot metal sample would be measured and transmitted to the big data platform at the end of the KR process. In other words, at a certain time before the BOF process, the real initial composition and

temperature could be obtained in real-time through dynamic data interaction technology. The component and temperature at the BOF terminal depended on the steel grade. According to the analysis of historical big data, the [C%] at the BOF terminal of a certain steel grade with different BOF modes was approximately normal, as shown in Figure 7 (sample number 7633). Therefore, The BOF terminal [C%] could be filled by the most probable component of the actual historical data for the same steel grade in the same BOF mode. In fact, in the historical data of the BOF, the real component and temperatures after the BOF process of 31 steel grades with three BOF modes were recorded in detail, which provides a rich data basis for supplementing the plan data on component and temperature.

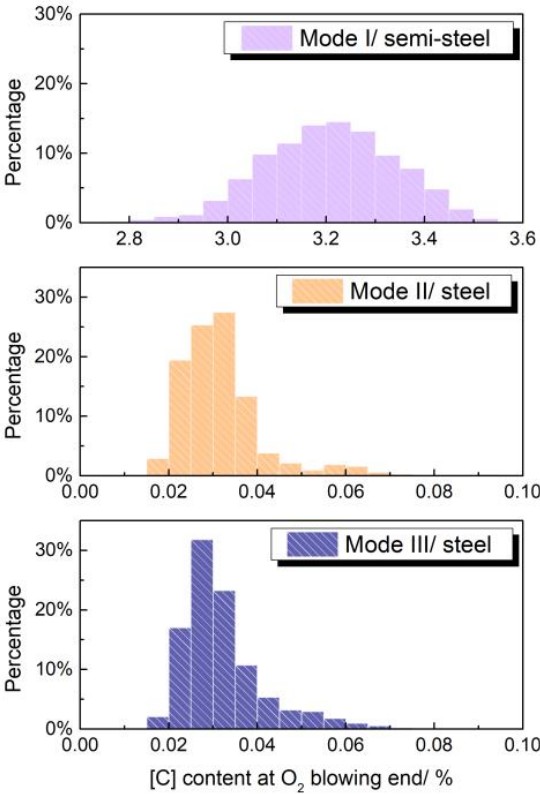

**Figure 7.** The distribution of [C%] at the BOF terminal for a steel grade.

### 2.3.2. Sample of Time Characteristic

At present, the plan data only consisted of the start time of the whole BOF process whiley the did not consider the operation time, the addition of scrap steel, addition of hot metal, and other operations before oxygen blowing. It is obviously unreasonable to directly regard the start time of the BOF process in the plan data as the start time of oxygen blowing. The time information of each BOF operation stage was recorded in detail in the historical performance data. As shown in Figure 8 (sample number: 55,887), according to whether scrap steel was added or not, the actual operation time before the oxygen blowing of the BOF was approximately normally distributed. Therefore, the operation time before oxygen blowing could be filled by the most probable period of the actual historical data, and then the BOF start time in the plan data could be modified to obtain the planned start and end time of oxygen blowing.

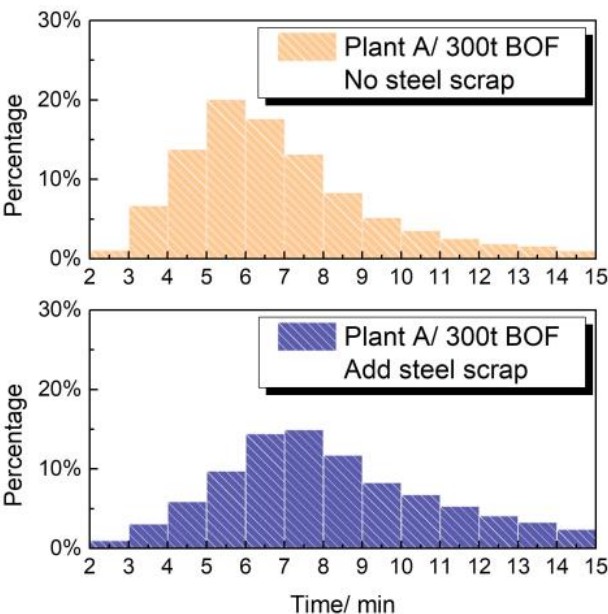

**Figure 8.** The time distribution before oxygen blowing.

### 2.3.3. Model Sample Building Based on BOF Plan Data

The original BOF plan data could be preprocessed according to the above principles. At the same time, for data anomalies, the z-score standardized method was adopted to normalize data distribution, and abnormal data outside the 3σ (three standard deviations) distribution was screened out. More than 40,000 model learning samples were obtained. Some original data and processing examples are shown in Tables 2 and 3. Due to the huge sample number, 2000 samples were randomly selected from the sample bank for each experiment, 80% of which were used as the training group and 20% as the test group. The test group was used to test whether overtraining occurred. When the prediction accuracy of the test group was basically the same as that of the training group, the model could be used.

**Table 2.** Original BOF plan data for model feature items.

| Plan ID | BOF ID | BOF Start Time | BOF End Time | Steel Grade | $M_{steel}$ |
|---------|--------|----------------|--------------|-------------|-------------|
| - | - | - | - | - | kg |
| 5028360 | 203B03848 | 15:57:56 | 16:24:56 | BN378001 | 288,832.00 |
| 5028363 | 202B03844 | 13:00:45 | 13:27:45 | BE05C011 | 287,436.00 |
| 5028378 | 203E04790 | 14:45:48 | 15:12:48 | BC09C011 | 309,581.00 |
| 5028379 | 203E04792 | 16:44:03 | 17:11:03 | BC06A001 | 297,670.00 |
| 5028380 | 201D04508 | 15:22:48 | 15:42:48 | BC06A001 | 297,670.00 |
| 5028380 | 202A04508 | 16:32:52 | 17:02:52 | BC06A001 | 297,670.00 |
| 5028381 | 203E04791 | 15:19:48 | 15:46:48 | AC064001 | 297,670.00 |
| 5028382 | 203E04793 | 17:20:57 | 17:47:57 | AC064001 | 297,670.00 |
| 5028383 | 203E04794 | 18:07:22 | 18:34:22 | AC064001 | 297,670.00 |
| 5028384 | 203A04509 | 18:44:24 | 19:11:24 | AC064001 | 297,670.00 |
| … | … | … | … | … | … |

**Table 3.** Processed samples of model feature items.

| Time Dynamic Variables | | | | Model Input Variables | | | | | | Model Output |
|---|---|---|---|---|---|---|---|---|---|---|
| Plan ID | *ost* | Duration | BOF Mode | De-[C] | De-[Si] | De-[P] | $m_{\text{hot metal/semi-steel}}$ | $T_{\text{hot metal/semi-steel}}$ | | $V_{O2}$ |
| - | - | - | - | % | % | % | t | °C | | m³ |
| 5028360 | 16:05:21 | 15.0 | 3 | 4.02 | 0.17 | - | 277.00 | - | | 12,954.00 |
| 5028363 | 13:06:05 | 11.7 | 2 | 3.17 | - | - | 301.00 | 1401 | | 10,861.00 |
| 5028378 | 14:53:28 | 15.5 | 3 | 4.78 | 0.47 | - | 286.00 | - | | 15,699.00 |
| 5028379 | 16:50:02 | 15.5 | 3 | 4.22 | 0.30 | - | 283.00 | - | | 14,286.00 |
| 5028380 | 15:29:58 | 8.5 | 1 | 1.08 | 0.42 | 0.04 | 309.11 | 1351 | | 3992.00 |
| 5028380 | 16:37:11 | 11.3 | 2 | 3.20 | 0.01 | - | 318.00 | 1392 | | 10,397.00 |
| 5028381 | 15:24:55 | 15.5 | 3 | 3.98 | 0.14 | - | 283.00 | - | | 14,499.00 |
| 5028382 | 17:28:12 | 15.5 | 3 | 4.19 | 0.15 | - | 285.00 | - | | 15,630.00 |
| 5028383 | 18:17:14 | 15.5 | 3 | 4.02 | 0.10 | - | 284.00 | - | | 14,181.00 |
| 5028384 | 18:50:33 | 15.5 | 3 | 4.09 | 0.12 | - | 280.00 | - | | 14,241.00 |
| … | | | … | … | … | | … | … | | … |

## 3. Results

### 3.1. Prediction Effect of Oxygen Consumption in a BOF

The prediction effect of oxygen consumption in a BOF was tested under a Chinese steelmaking plant for 2 days with a production capacity of about 58,000 tons. Figure 9 shows a comparison between the model prediction results and actual results. The diagonal line indicated that the predicted value was equal to the actual value. This meant that the closer the point was to the diagonal, the closer the prediction was to the actual values. The gray area indicated that the relative error was within ±5%. The predicted results were strictly distributed at 4000 m³, 11,000 m³, and 15,000 m³, followed by the three BOF modes. Most of the samples (100%, 99.9%, and 89.6% for the Mode I, II, and III, respectively) fell in the gray area, indicating that the probability of a model prediction error of less than 5% was more than 90%. The maximum average absolute error was about 450 m³. The lower the absolute oxygen consumption was, the higher the prediction accuracy was.

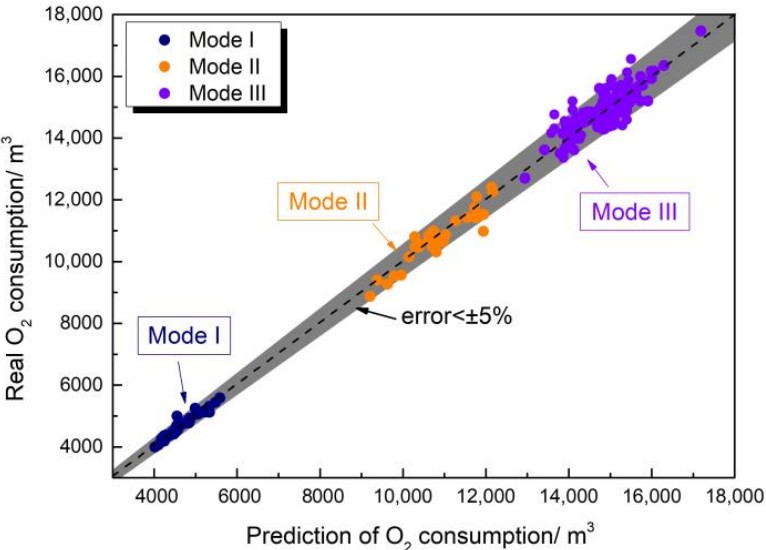

**Figure 9.** The prediction effect of oxygen consumption in a BOF.

### 3.2. Prediction Effect of Oxygen Consumption Flow in the Whole Steelmaking Process

According to the oxygen blowing time of BOF and the superposition plan of multi-furnace blowing, the predicted results of the total oxygen flow demand curve of the steelmaking process in the coming 8 h are shown in Figure 10. On the whole, the prediction results were in accordance with the real fluctuation tendency. However, the predicted curve was not in good agreement with the actual oxygen flow curve, and the mean absolute error (MAE) and root mean square error (RMSE) were as high as 577.98 $m^3$/min and 851.54 $m^3$/min, respectively. From the perspective of cumulative oxygen consumption, the absolute error of oxygen demand at the end was 22.5 $km^3$, and the relative error was only 3.05%, indicating that the model in this work still maintained high precision in predicting oxygen consumption volume.

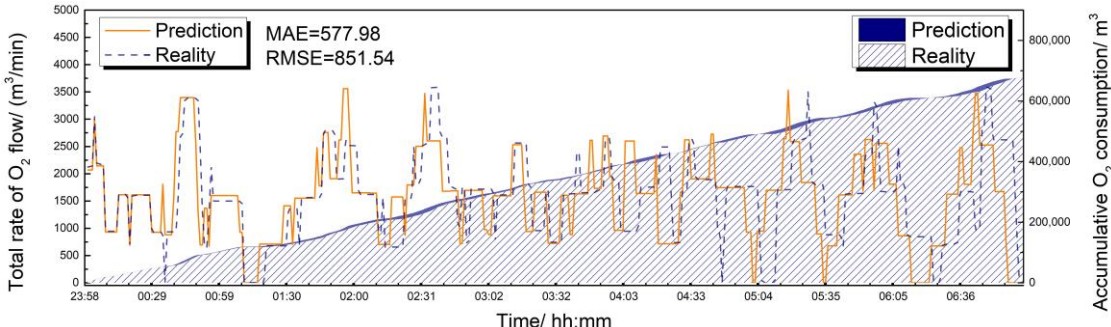

**Figure 10.** The prediction effect of the oxygen consumption flow rate according to the production plan in the next 8 h.

To analyze the reasons for the poor prediction accuracy of oxygen flow, the prediction effect in the next 0–2 h, 2–4 h, 4–6 h, and 6–8 h were compared, as shown in Figure 11. Figure 11a shows that the predicted curve had a good agreement with the real curve in the first 30 min, and then the real curve started to leave the predicted curve. Moreover, with the extension of the prediction period, the delay in the real curve became more obvious. In the period of 6–8 h, as shown in Figure 11d, the maximum delay time of the real curve was more than 10 min compared with that of the predicted curve, with a maximum MAE and RMSE of 965.99 $m^3$/min and 1195.78 $m^3$/min respectively. Therefore, it could be inferred that the time sequence mismatch between the predicted curve and the real curve was the direct cause of the poor accuracy of oxygen flow curve prediction. In fact, the model input samples in this work were the BOF plan data at the current moment. Within 30 min from the prediction trigger time to the future, the BOF plan was basically fixed, and the prediction accuracy of the model was high. However, in future, the BOF plan would be adjusted due to the disturbance of the seelmaking process, that is, dynamic scheduling, which led to an advance or delay in the BOF plan start/end time. Therefore, the dynamic scheduling of BOF plan was the key cause of the poor accuracy in oxygen flow curve prediction. Nevertheless, the time from oxygen pipe network to BOF was only about 2~3 min. During dynamic operation, the model in this work could accurately predict the future oxygen flow demand 30 min in advance, which had the potential to maintain the dynamic balance of oxygen with short-term supply.

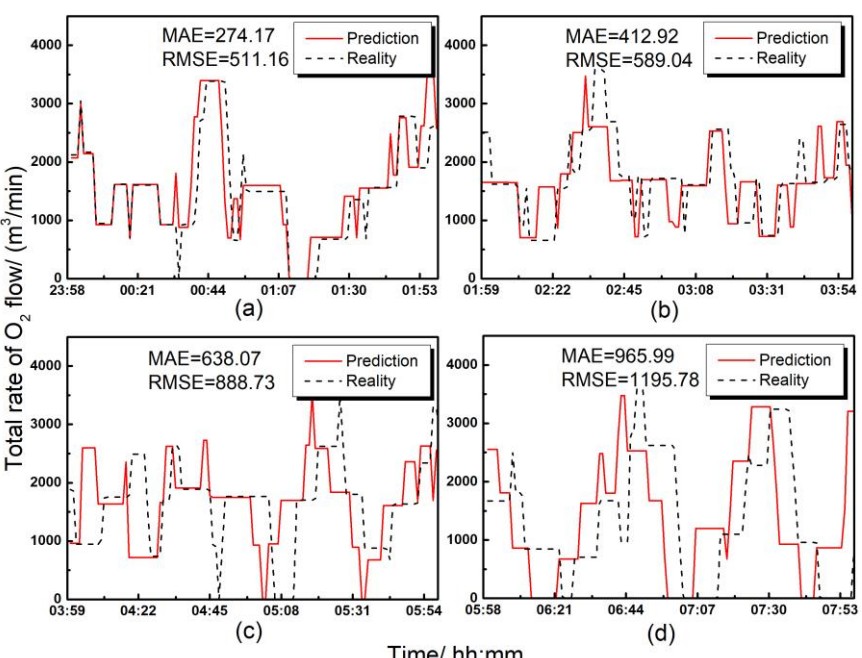

**Figure 11.** The prediction effect of the oxygen consumption flow at different time periods in the future: (**a**) 0~2 h; (**b**) 2~4 h; (**c**) 4~6 h; (**d**) 6~8 h.

## 4. Discussion

A dynamic prediction model of BOF process has been reported in many studies. Most of them focused on the accurate tracking of the BOF process and the dynamic optimization of the operation control. The oxygen demand is only a boundary condition in the dynamic control model, which can be dynamically modified but cannot be predicted in advance, and provides decision support for oxygen scheduling. The Industry 4.0 strategy requires the steel industry to transform to low carbon. Establishing an accurate prediction model and realizing the dynamic balance of oxygen is a key measure to strengthen the production and energy management in iron and steel enterprises under the situation of overall energy savings and consumption reductions. In addition, most of the current oxygen prediction models took historical production performance data as input data, which could be evaluated afterward, but they were difficult to predict before production and struggled to meet the requirements for the dynamic prediction of oxygen.

Therefore, the oxygen consumption mechanism, model algorithm, and other aspects were reasonably simplified in this work, and the industrial production plan data with less information were supplemented and improved. Based on the dynamic database, the model-running framework of prediction, clock, and training multithreading was designed to improve the model efficiency and realize the dynamic prediction of oxygen demand.

In the aspect of production plan data preprocess, a total of eight variables in three categories were determined by combining oxygen consumption mechanism and industrial big data characteristics. Among them, variables such as steel composition and temperature after the end of blow, as well as oxygen-blowing time, were not included in the production plan data. Since the composition, temperature, and oxygen-blowing time were related to steel grade and BOF modes, this work was based on historical production performance data. The end composition distribution, temperature distribution, time distribution, and oxygen-blowing duration distribution of each steel grade were analyzed under different BOF modes. The historical data distribution of the steel in this mode was used to supplement the missing key variables of the model sample in the production plan data. The results revealed that the average relative error of the prediction based on the production plan data was less than 5%, which could satisfy the precision demands of oxygen consumption prediction in industrial production. This proved the feasibility of modeling based on production plan data and the rationality of the data-filling method used in this work.

Under the premise that the production plan was basically fixed (within 30 min), the model in this work maintained a good prediction accuracy. However, with the scheduling of the production plan, the prediction accuracy obviously decreased, which indicated that the production plan was eventually executed or not was the limiting factor of the prediction accuracy in this work. The dynamic scheduling of the production plan was inevitable in the actual BOF production process, which meant that the production plan data input to the model must be refreshed in real-time. Therefore, in model framework design, in addition to the conventional model training and prediction thread, the model clock thread synchronized with the dynamic database was specifically developed. At the same time, this thread was also set to store information such as the real clock, plan change flag, prediction and training trigger flag. Once the production plan was changed, the model was run and output the result. The last result was overwritten to dynamically correct the model prediction result. Under the experimental conditions used in this work, the dynamic prediction results of the model could accurately reflect the fluctuation of oxygen flow demand in the next 30 min, which could provide a scheduling basis and support for the dynamic balance of oxygen in the production-energy system, thus reducing the pressure fluctuations and oxygen release of the oxygen pipe network, saving energy and reducing consumption from the perspective of the system.

## 5. Conclusions

In this work, through an analysis of the reaction mechanism and industrial big data, it was determined that duplex dephosphorization, duplex decarbonization, and traditional mode were affected by material, operation, target, function, etc., and their oxygen consumption characteristics were different. Because of the lack of key information such as composition and temperature in the plan data, the dynamic prediction model samples for different modes were established based on historical big data mining.

Based on the industrial data simulation from a Chinese steelmaking plant, the results revealed that the average relative error of the model was less than 5% and the maximum average absolute error was about 450 $m^3$, which could meet the accuracy requirements regarding the static balance in the long-term prediction of the BOF oxygen consumption.

The dynamic adjustment of the actual production plan was an important factor affecting the predict accuracy of the oxygen consumption flow. In this work, a time sequence-matching control thread based on a dynamic database was developed to synchronize the production plan schedule in real-time and dynamically revise the model prediction results. Under the experimental conditions in this Chinese steelmaking plant, the model could accurately predict the BOF oxygen flow demand 30 min in advance, which could provide a scheduling basis and support for the short-term oxygen demand plan and the dynamic balance in the production-energy system.

**Author Contributions:** Conceptualization, L.Z. and Y.L.; Investigation, K.Z., L.Z., Y.L. and S.C.; Data curation, S.C.; Writing—original draft, K.Z.; Writing—review & editing, K.Z. and Z.Z.; Project administration, K.Z. and Z.Z. All authors have read and agreed to the published version of the manuscript.

**Funding:** This research was funded by the National Key R&D Program of China (No. 2020YFB1712803), the National Key R&D Program of China (No. 2021YFE0113200), and the Chongqing Postdoctoral Science Foundation project (No. cstc2020jcyj-bshX0104).

**Data Availability Statement:** Due to the commercial restrictions, the data that support the findings of this study are available from the corresponding author, Prof. Zheng Z., upon reasonable request.

**Acknowledgments:** We thank Shougang Jingtang United Iron and Steel Co., Ltd. for making the data collection for this work. We are also grateful to Liu J. and He Y. for helpful discussions and valuable suggestions.

**Conflicts of Interest:** The authors declare no conflict of interest.

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
