# Peer review of "Method for Dynamic Prediction of Oxygen Demand in Steelmaking Process Based on BOF Technology"

_processes, doi:10.3390/pr11082404_

Round 1

Reviewer 1 Report

While the topic is interesting most of the presented results hide what the authors did, or hide the essence of the work.

The authors need to improve the paper in such a way that the starting and end content, as well as their change, of C, P and Si are clearly visible. The changes must be correlated to the blown O2 and temperature (start, end, and change of temperature). Furthermore, the authors must present a clear relation of dephosphorization to the temperature, C, Si, as well as FeO and CaO in the slag.

The only mistake that consistently poped up was "et al" instead of "et al.".

Author Response

Thank reviewer for your valuable comments. These comments are very useful to our work. Your comments and our responses are as following:

Comments and Suggestions for Authors:While the topic is interesting most of the presented results hide what the authors did, or hide the essence of the work.

The authors need to improve the paper in such a way that the starting and end content, as well as their change, of C, P and Si are clearly visible. The changes must be correlated to the blown O2 and temperature (start, end, and change of temperature). Furthermore, the authors must present a clear relation of dephosphorization to the temperature, C, Si, as well as FeO and CaO in the slag.

Response: Thanks for your comments. We have added a Figure to illustrate the difference in oxygen consumption between BOF modes in the revised manuscript at line 184. In addition, the oxidation of carbon is mainly responsible for oxygen consumption, and the absolute amount of oxidized carbon is related to the starting and ending carbon content and the weight of hot metal. Therefore, we used XYZ plots to represent the coupling of oxygen consumption - oxidized carbon content - hot metal weight. However, it is difficult to express this relationship using the starting and ending carbon content.

As for the dephosphorization, our analysis is as follows: [P] is mainly oxidized by (FeO) and solidified by (CaO) as calcium phosphate into the slag, and this reaction is an exothermic reaction. Therefore, (FeO), (CaO), and low temperature (about 1400 ℃) are all favorable factors for dephosphorization. The oxidization of [Si] result in low basicity of the slag, so [Si] is not conducive to dephosphorization. as the focus of this work was on the consumption of oxygen in the BOF, rather than discussing the related influencing factors of dephosphorization, the composition of slag was not concerned. The relationship between decarbonization and dephosphorization is complicated. On the one hand, the decarbonization reaction will consume (FeO) and release heat, which is not conducive to the thermodynamic conditions of dephosphorization. On the other hand, the decarburization reaction is conducive to the melting of slag and the stirring of molten steel, which is conducive to the dynamic conditions of dephosphorization. Your question makes us to realize the absence of this key part. And we supplement the above analysis with the BOF model in the revised manuscript at line 198-line 204.

The only mistake that consistently poped up was "et al" instead of "et al."

Response: Thanks for your correction. We have made a comprehensive correction to this mistake in the revised manuscript.

Reviewer 2 Report

The authors have employed a combination of mechanism analysis and industrial big data to investigate the characteristic factors in steel making process that affecting oxygen consumption in different modes of the Basic Oxygen Furnace (BOF), such as duplex dephosphorization, duplex decarbonization, and traditional modes. Building upon these findings, the authors has proposed a dynamic prediction modeling method for BOF oxygen demand, considering mode classification. To address the unique characteristics of BOF production, the manuscript has developed a series of matching control modules based on dynamic adaptations of the production plan. These modules facilitated recalculations of the model predictions as per the specific production organization of the BOF. Through simulation tests using industrial data, the model demonstrated an average relative error of less than 5% for each BOF mode, with a mean absolute error of approximately 450m3. Their results revealed that the model successfully achieved an accurate prediction of dynamic oxygen demand with a lead time of 30 minutes.

These results offer valuable methodological support and a foundation for long-term demand planning to achieve static balance, as well as for short-term real-time scheduling to maintain dynamic oxygen balance in steelmaking. The developed dynamic prediction modeling method, along with the corresponding control modules, contributes to improving the overall efficiency and effectiveness of oxygen utilization in the steelmaking process.

The manuscript is well written and organized and shows interesting results and discussion points and it may have some originality, and application for the steel industry. Below is a summary list of suggestive minor revisions that might help improve the manuscript.

Author Response

Response Letter

Thank reviewer for your valuable comments. These comments are very useful to our work. Your comments and our responses are as following:

Comments and Suggestions for Authors:The authors have employed a combination of mechanism analysis and industrial big data to investigate the characteristic factors in steel making process that affecting oxygen consumption in different modes of the Basic Oxygen Furnace (BOF), such as duplex dephosphorization, duplex decarbonization, and traditional modes. Building upon these findings, the authors has proposed a dynamic prediction modeling method for BOF oxygen demand, considering mode classification. To address the unique characteristics of BOF production, the manuscript has developed a series of matching control modules based on dynamic adaptations of the production plan. These modules facilitated recalculations of the model predictions as per the specific production organization of the BOF. Through simulation tests using industrial data, the model demonstrated an average relative error of less than 5% for each BOF mode, with a mean absolute error of approximately 450m3. Their results revealed that the model successfully achieved an accurate prediction of dynamic oxygen demand with a lead time of 30 minutes.

These results offer valuable methodological support and a foundation for long-term demand planning to achieve static balance, as well as for short-term real-time scheduling to maintain dynamic oxygen balance in steelmaking. The developed dynamic prediction modeling method, along with the corresponding control modules, contributes to improving the overall efficiency and effectiveness of oxygen utilization in the steelmaking process.

The manuscript is well written and organized and shows interesting results and discussion points and it may have some originality, and application for the steel industry. Below is a summary list of suggestive minor revisions that might help improve the manuscript.

Response: Thanks for your appreciation. However, we could not find the summary list in your review report. We are looking forward to your suggestions to improve our manuscript.

Reviewer 3 Report

The article titled "Dynamic prediction method of oxygen demand in steelmaking process considering BOF mode classification" is proposed to be investigated for possible publication in the Journal of Processes. Authors employed simulations and big data for analyzing the process mechanism based on the oxygen demand in the way of industrialization of the steelmaking process considering BOF mode classification. Although the article is organized in a correct way and different aspects of the process are covered, there are some hints that can increase the quality of this manuscript as below:

1- Abstract: define the BOF abbreviation the first time it is used.
2- "[email protected](K.Z.); [email protected](L.Z); [email protected](Y.L.); [email protected](S.C)" These are emails? Are they correct?!
3- It is good to give a look at the recently published article very related to the current work:
A Novel Dynamic Operation Optimization Method Based on Multiobjective Deep Reinforcement Learning for Steelmaking Process
DOI: 10.1109/TNNLS.2023.3244945

Please describe your novelty and your working distance from the published work. Is the current research could not be performed by the method and procedure published in the mentioned articles?

4- BOF should be defined at least the first time you use it (line 27).

5- The first paragraph of the introduction section is not a good start for a research paper. More details about the process and BOF is actually needed for a section named "Introduction". The second paragraph is also somehow a copy of the first paragraph!

6- The literature review part is not comprehensive enough. Moreover, the references are not well-fitted to the subject and are not up to date.

7- The balances are well-known and no need for too much description at least for the first equation.

8- "... ki is the thermodynamic and kinetic correlation coefficients..." How did you use a correlation coefficient for both critical aspects of thermodynamics and kinetics?

9- "In gernral, they are generally evaluated by experience." Do you mean by experiments? Even the word "general" has a typo. What you proposed is not correct by the way.

10- What is the exact purpose of Figure 1? It is not well-described in the body and the title.

11- Figure 2 is not high quality; is it prepared by authors or it is adopted?

12- What is brought as the novelty of this model, especially in the model theory?

13- The same as cm11 is for Figure 4. Also, what is the main novelty and the logic of the algorithm which is new?

14- Same for Figure 5. Is it adopted? what is the reference?

Author Response

Response Letter

Thank reviewer for your valuable comments. These comments are very useful to our work. Your comments and our responses are as following:

Comments and Suggestions for Authors:

1.Abstract: define the BOF abbreviation the first time it is used.

Response: We have added the explain of abbreviation BOF in the abstract at line 14 in revised manuscript.

2."[email protected](K.Z.); [email protected](L.Z); [email protected](Y.L.); [email protected](S.C)" These are emails? Are they correct?!

Response: Thanks for your mention. After the verification, we determined that they were all correct and valid email addresses.

  1. It is good to give a look at the recently published article very related to the current work:

A Novel Dynamic Operation Optimization Method Based on Multiobjective Deep Reinforcement Learning for Steelmaking Process

DOI: 10.1109/TNNLS.2023.3244945

Please describe your novelty and your working distance from the published work. Is the current research could not be performed by the method and procedure published in the mentioned articles?

Response: Thanks for your recommendation. The ideas and modeling methods of this published article are very inspiring for our work, and we cite it in the introduction. Especially in the construction of energy function by physical heat, reaction heat, and consumption heat of molten steel, and then coupling to the model and algorithm. This is an effective mechanical-data fusion modeling method.

In the current work, we concentrated on predicting oxygen demand of BOF on a macro level to provide decision support for oxygen network scheduling, so as to achieve the balance between supply and demand of oxygen system. The literature you recommend focused on the accurate tracking of the BOF process and the dynamic optimization of the operation control. The difference of modeling goals leaded to our working distance from the published work.

In addition, the data sample of the model in the literature is historical production data, while the data sample of the model in this work is historical plan data. The above two types of data contain much different information. Therefore, one of the novelties of this work was to fill in the BOF plan data by analyzing the informative historical production data, so as to realize the oxygen prediction of the BOF before production. Moreover, different characteristic factors were selected as the model input in this work according to the BOF mode to improve the prediction accuracy. Due to the differences in the above model purposes, training data, and logistic, the current research could not be directly performed by the method and procedure published in the mentioned articles. Nevertheless, the model in your recommended literature has advantages in algorithm optimization, prediction accuracy, mechanical-data fusion, etc., which provided a good reference for our further research on oxygen accurate prediction and scheduling decision.

  1. BOF should be defined at least the first time you use it (line 27).

Response: We have added the explain of abbreviation BOF in the revised manuscript at line 32.

  1. The first paragraph of the introduction section is not a good start for a research paper. More details about the process and BOF is actually needed for a section named "Introduction". The second paragraph is also somehow a copy of the first paragraph!

Response: Thanks for your comments. We are also realized that this part of the content is not appropriate. Therefore, we added more details as well as relevant literatures, and reorganized the content and logic of this section in revised manuscript. Line 28-line 53.

  1. The literature review part is not comprehensive enough. Moreover, the references are not well-fitted to the subject and are not up to date.

Response: Same as the comment 5, we reorganized the literature review based on your suggestions and updated the relevant literature in revised manuscript, especially for the background. (Line 28-line 53, Line 70-73. Line 474-481)

  1. The balances are well-known and no need for too much description at least for the first equation.

Response: Thanks for your suggestion and we accept it. The purpose of our initial description was to illustrate the relationship of oxygen consumption in the BOF to the non-metallurgical reader. Obviously, the balances are simple and easy to understand for metallurgical researchers. In the introduction section, we have described the more details about BOF according to comment 6. Therefore, we have simplified the description of this section and deleted the first equation. (Line 118-line 123)

  1. "... ki is the thermodynamic and kinetic correlation coefficients..." How did you use a correlation coefficient for both critical aspects of thermodynamics and kinetics?

Response: Thanks for your critical question. We agree that theoretically, the extent of oxygen reaction can be calculated through thermodynamic activity coefficients, and the reaction rate such as blowing oxygen for decarbonization can be calculated through kinetic mass transfer coefficients. In industrial practice, however, the reactions in BOF process are very complex, which leads to the need to modify the theoretical coefficients of thermodynamics and kinetics, and the correction method is extremely complex. The coefficient ki in this work is a simplified expression of the complex relationship between each element of hot metal and oxygen consumption. We admit that the expression of “thermodynamic and kinetic correlation coefficients” was not proper, so we modify it: ki is the simplified expression of a series of coefficients of the complex relationship between each element of hot metal and oxygen consumption. So is kj. (Line 139-146)

  1. "In gernral, they are generally evaluated by experience." Do you mean by experiments? Even the word "general" has a typo. What you proposed is not correct by the way.

Response: Thanks for your correction. What we want to express is that some accurate parameters or correction coefficients under a limited number of typical conditions can be obtained through experiment or measurement. But in the industrial practice, more of them are obtained by the production operator summing up the historical production experience and repeatedly adjusting and optimizing the process. Your question reminds us that this sentence is not appropriate here, so we have deleted it. (Line 139-146)

  1. What is the exact purpose of Figure 1? It is not well-described in the body and the title.

Response: Thanks for your correction. We have optimized the representation of the Figures. A Figure to illustrate the difference in oxygen consumption between BOF modes was added as Figure 1 in the revised manuscript. In addition, the oxidation of carbon is mainly responsible for oxygen consumption, and the absolute amount of oxidized carbon is related to the starting and ending carbon content and the weight of hot metal. Therefore, we used XYZ plots to represent the coupling of oxygen consumption - oxidized carbon content - hot metal weight.

  1. Figure 2 is not high quality; is it prepared by authors or it is adopted?

Response: We have improved the quality of Figure 2.

12- What is brought as the novelty of this model, especially in the model theory?

Response: Thanks for your question. We believe the model has the following novelty. At present, many oxygen models are based on after-production data, with is accurate and abundant. These models are good at analysis after production while cannot predict in advance. The model in this work was based on the plan data, which is usually worked out one shift group (about 8h) in advance. Therefore, the model in this work can theoretically predict the oxygen demand of the BOF 8h in advance, thus providing decision-making support for oxygen scheduling, balancing the pressure of the oxygen network, and reducing oxygen release.

Without doubt, the plan of the BOF in actual production is dynamically adjusted with various disturbances. Normally, the further away from the production time, the greater the adjustment of plan. This directly affect the accuracy of oxygen flow prediction. As a result, the model in this work designed multiple threads to implement real-time predictions based on plan data, iteratively correcting the prediction results.

In addition, the plan data before production generally contains very little information about the data for model learning samples. Therefore, a plan data imputation method based on historical data analysis and mining was proposed in this work to it is necessary to preprocess the production plan data to obtain reliable model learning samples.

  1. The same as cm11 is for Figure 4. Also, what is the main novelty and the logic of the algorithm which is new?

Response: Thanks for your question. The algorithm used in this work was relatively simple, that is, BP neural network algorithm. The novelty we believe is to establish neural network models with different characteristic variables according to the BOF mode. The algorithm running logic is: First, the BOF mode is determined according to plan data; then the corresponding characteristic variables are extracted according to the BOF mode as the input of the BP neural network model, and output the prediction of the oxygen consumption of the furnace; finally, the oxygen consumption of the furnace is converted into the oxygen flow rate according to formulas 5~8. We also supplement the above description of the algorithm logic in the revised manuscript.

  1. Same for Figure 5. Is it adopted? what is the reference?

Response: Thanks for your question. We have improved the quality of Figure 5. As for the reference, we are not quite sure if you are referring to some literatures. The dynamic operation framework of the model was designed and developed by ourselves. Therefore, no references are available. 

Reviewer 4 Report

Review: processes: paper no. 2497980 12 July 2023

Dear Authors,

 The topic of the paper is important for the Journal "Processes" but you need to improve the paper, I have some comments, as an expert on metallurgy in Central European countries:

Note 1: Now is:

Dynamic prediction method of oxygen demand in steelmaking process considering BOF mode classification

In my opinion it should be: Dynamic prediction method of oxygen demand in steelmaking process based on BOF technology

Note. 2. Sentence in Introduction "The steel output of BOF accounts for more than 80% of the total output of steelmaking" (27) my question is about 80%: when (year) and where (world, country...).

In the first sentence or in the keywords, the authors need to explain the abbreviation BOF, not every reader knows the abbreviations of steelmaking technology.

After references 7and 8, I would give more statistical information (data from WorldSteel ) about BOF technology , as well as more about the directions of change related to the decarbonisation of metallurgy and the implementation of the Industry 4.0 concept in mills (steel Industry) ( https://doi.org/10.3390/en14113034). BOF technology will be phased out in Europe (EU countries) because the EU is pursuing a net zero policy.

Please have the authors write more about the European steel industry (Industry 4.0 in the European Iron and Steel Industry: Towards an Overview of Implementations and Perspectives (fraunhofer.de). Available online: https://www.isi.fraunhofer.de/content/dam/isi/dokumente/cce/2018/Industry-4-0-Implementation-andPerspectives_Steel-Industry_Working%20document.pdf).

Note 3: In the Introduction (at the end) the authors should formulate the purpose of the research.

The paper does not have a literature review; if other reviewers write such a note, I will add one.

Note 4: Methods (2) Please change to Materails and methods. In such a section , the authors should describe how the study was performed (drawing, schematic), present data on the study input, and then describe: 2.1, 2.2. etc.

Note 5: Underneath the figures should be sources: own research.

I have no comments on the results, although before para. 3.1. the authors could write how they present the results (describe segments 3.1, 3.2) and why only in this arrangement the results are analysed.

The discussion will change if the authors add a chapter: Literature review. This is because there is a principle that the discussion should refer to one's own and others' research.

Note 6. Not enough literature because since there is no literature review the references are short. I encourage the authors to make a section, after Introduction, Literature review, or add more text to the Introduction with references (new papers on I 4.0, decrabonisation and other problmes in steel Industry (technology BOF).

Best wishes

Reviewer

Author Response

Response Letter

Thank reviewer for your valuable comments. These comments are very useful to our work. Your comments and our responses are as following:

Comments and Suggestions for Authors:

Note 1: Now is: “Dynamic prediction method of oxygen demand in steelmaking process considering BOF mode classification”. In my opinion it should be: “Dynamic prediction method of oxygen demand in steelmaking process based on BOF technology”

Response: Thanks for your suggestion. We intended to highlight the difference between duplex dephosphorization, duplex decarbonization, and traditional mode Your suggestion makes us realize that they are BOF technology. So, your suggestion is more appropriate for this work. We have changed it as “Dynamic prediction method of oxygen demand in steelmaking process based on BOF technology” in the revised manuscript. (Line 2)

Note. 2. Sentence in Introduction "The steel output of BOF accounts for more than 80% of the total output of steelmaking" (27) my question is about 80%: when (year) and where (world, country...).

In the first sentence or in the keywords, the authors need to explain the abbreviation BOF, not every reader knows the abbreviations of steelmaking technology.

After references 7and 8, I would give more statistical information (data from WorldSteel ) about BOF technology , as well as more about the directions of change related to the decarbonisation of metallurgy and the implementation of the Industry 4.0 concept in mills (steel Industry) ( https://doi.org/10.3390/en14113034). BOF technology will be phased out in Europe (EU countries) because the EU is pursuing a net zero policy.

Please have the authors write more about the European steel industry (Industry 4.0 in the European Iron and Steel Industry: Towards an Overview of Implementations and Perspectives (fraunhofer.de). Available online: https://www.isi.fraunhofer.de/content/dam/isi/dokumente/cce/2018/Industry-4-0-Implementation-andPerspectives_Steel-Industry_Working%20document.pdf).

Response: Thanks for your question. For 80% and abbreviation BOF, we reviewed the data from World Steel Association and updated it as follows: the Basic Oxygen Furance (BOF) is responsible for approximately 70% of the steel production worldwide in 2021.

For the European steel industry, we have carefully read the references you provided. However, the link of “Industry 4.0 in the European Iron and Steel Industry” is out of work. So we checked the other relevant reports. We understand that with the proposal of Industry 4.0, the European steel industry is moving towards low-carbon and intelligent development. Related technologies such as hydrogen metallurgy, direct reduction, electric arc furnace, and digitization are also the focus. However, in many developing countries, such as China, the BF-BOF route still exists in large numbers and is difficult to be replaced by new technologies in the short term. Therefore, how to realize energy saving based on the existing BF-BOF route is also a direction that need to be studied. Accordingly, the prediction method of oxygen demand of BOF was studied in this work to provide decision-making support for oxygen scheduling, achieve oxygen supply and demand balance, reduce waste. The references you recommended are important revelation for the background of our work and we have added it to the Introduction in the revised manuscript. (Line 28-33)

Note 3: In the Introduction (at the end) the authors should formulate the purpose of the research.

The paper does not have a literature review; if other reviewers write such a note, I will add one.

Response: Thanks for your suggestion. We have added the purpose of our work at the end of the Introduction. However, this manuscript an article and not a review. According to the structural requirements of the journal, the literature review cannot be listed separately, so it can only be integrated in the Introduction. (Line 97-105)

Note 4: Methods (2) Please change to Materails and methods. In such a section , the authors should describe how the study was performed (drawing, schematic), present data on the study input, and then describe: 2.1, 2.2. etc.

Response: Thanks for your suggestion. We have revised and added an overall description of the research method and data on study input. (Line 106-115)

Note 5: Underneath the figures should be sources: own research.

I have no comments on the results, although before para. 3.1. the authors could write how they present the results (describe segments 3.1, 3.2) and why only in this arrangement the results are analysed.

The discussion will change if the authors add a chapter: Literature review. This is because there is a principle that the discussion should refer to one's own and others' research.

Response: Thanks for your question. For the results, we assume that you want to ask why we evaluate prediction accuracy based on BOF mode and time periods in the future. One of the novelties we believe in this work is to establish neural network models with different characteristic variables according to the BOF mode. Therefore, we compared the oxygen consumption calculated by the model under different BOF modes with the reality, as well as to verify the prediction accuracy of the model under different BOF modes. In addition, a steelmaking plants generally has several BOF, which may work at the same time. This condition led to a surge in oxygen demand. Therefore, we also compared the predicted and reality of oxygen flow at different time periods in the future, and verified the support of the proposed model for oxygen supply scheduling.

For discussion part, your question makes us aware of the inadequacy of the comparison between our research and that of others. We added this content in the front of this section. (396-407)

Note 6. Not enough literature because since there is no literature review the references are short. I encourage the authors to make a section, after Introduction, Literature review, or add more text to the Introduction with references (new papers on I 4.0, decrabonisation and other problmes in steel Industry (technology BOF).

Response: Thanks for your suggestion. We have added references and expanded the Introduction, especially on the background of Industry 4.0 and decarbonization. (Line 474-481)

Round 2

Reviewer 3 Report

Thanks for the efforts that may be done. My concerns are tried to be addressed and the manuscript seems publishable as is.